# Utility of Contrast-Enhanced Ultrasound in Optimizing Hepatic Abscess Treatment and Monitoring

**DOI:** 10.3390/jcm13175046

**Published:** 2024-08-26

**Authors:** Adam Dobek, Mateusz Kobierecki, Konrad Kosztowny, Oliwia Grząsiak, Adam Fabisiak, Krzysztof Falenta, Ludomir Stefańczyk

**Affiliations:** 1Department of Radiology and Diagnostic Imaging, Norbert Barlicki Memorial Teaching Hospital No. 1, Medical University of Lodz, 90-153 Lodz, Poland; 2Department of Diagnostic Imaging, Polish Mother’s Memorial Hospital Research Institite, 90-153 Lodz, Poland; 3Department of General Surgery and Transplantology, Norbert Barlicki Memorial Teaching Hospital No. 1, Medical University of Lodz, 90-153 Lodz, Poland; 4Department of Digestive Tract Diseases, Norbert Barlicki Memorial Teaching Hospital No. 1, Medical University of Lodz, 90-153 Lodz, Poland

**Keywords:** abscess, contrast-enhanced ultrasound, liver, percutaneous drainage, ultrasound

## Abstract

**Background/Objectives**: Untreated hepatic abscesses (HAs) have an 80% mortality rate and can be caused by bacteria and fungi. Previously managed with surgery, current treatments now utilize interventional radiology and antibiotics, reducing complications to 2.5%. This study evaluates contrast-enhanced ultrasound (CEUS) for better drainage placement and monitoring, overcoming conventional ultrasound’s limitations in detecting the HA liquefied portion. **Methods**: We conducted a retrospective study of 50 patients with HAs confirmed via computed tomography (CT) scans. Inclusion criteria comprised specific clinical symptoms and laboratory parameters. Both B-mode and CEUS were utilized for initial and follow-up imaging. **Results**: In the CEUS studies, the mean size of HAs was 6.26 cm, with pus displaying significantly lower echogenicity compared to the HA pouch and liver parenchyma in all phases. Classification by size (>6 cm, <6 cm) and volume (>113 mL, <113 mL) revealed differences in the assessment of fluid volume between CEUS and B-mode. **Conclusions**: CEUS is valuable for diagnosing, performing therapeutic procedures, and monitoring HA. It provides precise real-time assessment of HA morphology, including dimensions and volume. If the liquefied volume of an HA exceeds 113 mL, it may qualify for drainage placement. CEUS can replace CT as an effective, less harmful, and cheaper method, eliminating the need for multiple radiological departments. While CEUS is a safer, cost-effective alternative to CT for HA evaluation and monitoring, comprehensive clinical evaluation remains essential. Therefore, CEUS should be part of a broader diagnostic and monitoring strategy, not a stand-alone solution.

## 1. Introduction

The mortality rate for untreated hepatic abscess (HA) is 80% [1,2]. The etiology of HA can vary and includes bacteria, fungi, and parasites. Etiology can also be mixed. Historically, HA was primarily treated with surgical drainage, which often led to various complications. Today, interventional radiology methods, along with targeted antibiotic therapy, are considered the gold standard in the treatment of HA. This path of treatment is far more comfortable and acceptable for patients, and the convalescence period is shorter, with a complication ratio lowered to 2.5% [1,3]. The most common way of drainage insertion is under ultrasonography guidance. It allows real-time assessment of drain location during the procedure and choosing the best approach for the operator. Nonetheless, HA can present variously and can be misdiagnosed as a necrotic tumor, hematoma, or cyst [1]. Moreover, HA, depending on the etiological factor and the duration, can present variously in four morphological forms: I—tumor-like; II—honeycomb; III—lacunar; IV—cystic-like [4]. The liquefied part of the abscess may not be visible or underestimated in B-mode ultrasound (B-mode), which can pose difficulties for the operator in choosing the optimal drain placement. This study aimed to evaluate the effectiveness of contrast-enhanced ultrasound (CEUS) in analyzing the liquid component of HA. The goal was to determine the optimal placement of catheters within these lesions and to facilitate ongoing monitoring of the drainage process.

## 2. Materials and Methods

This research adhered to the Declaration of Helsinki guidelines and received approval from the Bioethics Committee (RNN/266/22/KE). Participants in the study gave informed consent. Inclusion criteria required a definitive diagnosis of HA via computed tomography (CT) scans, GE Healthcare Revolution CT and GE Lightspeed VCT 64 Slice scanners were utilized (GE Healthcare, Milwaukee, WI, USA), identified by hypodense lesions with peripheral enhancement and central areas lacking enhancement, suggestive of pus. Other observed characteristics included irregular boundaries of the HA and gas presence within these cavities. The study also took clinical symptoms into account, such as upper right quadrant pain, jaundice, tenderness, fever, nausea, weight loss, and night sweats. Comprehensive hepatic function assessment was supported by measuring AST and ALT levels, aiding in both the diagnosis and the ongoing monitoring of HA. Each patient underwent at least one multiphase contrast-enhanced CT scan to confirm each lesion throughout their treatment. Exclusion criteria aligned with the contraindications for using contrast in ultrasound as per SonoVue (Bracco Imaging S.p.A., Milan, Italy) guidelines, including respiratory insufficiency, acute coronary syndrome, adverse reactions to contrast media, or pregnancy. Cut-off values for HA dimensions (6 cm) and volume (113 mL) were derived from a meta-analysis by Lin et al., providing a basis for differentiating between treatment approaches [3].

### 2.1. Patients

The retrospective analysis included 50 patients—23 females and 27 males—with an average median age of 66 years (IQR 53–75 years, range 21–97 years, mean (SD) 65 (10) years), who had been diagnosed with HA. A total of 10 patients had multiple lesions, ranging from 2 to 9 HA, with a total of 112 lesions. The study included initial B-mode and CEUS examinations, as well as potential follow-up. The size, volume of the liquefied part, and morphology type of HA, according to the adapted methodology, were independently evaluated via B-mode and via CEUS in the post-contrast images. In addition, the group was divided into lesions measuring >6 cm (53 lesions) and those measuring <6 cm (59 lesions) [3,4]. HAs were confirmed via CT based on the presence of hypodense lesions with peripheral enhancement and central nonenhancing areas, indicative of pus. Additional findings included irregular HA borders and the presence of gas within the HA cavity. In our study, no cases of sepsis or sudden onset of shock were observed. All patients exhibited at least one of the following clinical symptoms: pain in the right upper quadrant, jaundice, tenderness, fever, nausea, weight loss, or night sweats. There were no asymptomatic cases included in the study. A total of 39 patients were qualified for percutaneous drainage (PCD), while 11 were disqualified due to clinical status or inaccessible HA location. A total of 28 patients underwent CEUS follow-up during the drainage process. A total of 11 patients who were disqualified from PCD due to clinical status or inaccessible HA locations were treated exclusively with targeted antibiotic therapy. For the 28 patients who underwent PCD, CEUS follow-up was performed at 7-day intervals, or earlier if the discharge of pus from the drain ceased, to monitor the progression and resolution of the HA. For the remaining 22 patients, CEUS was performed only once at admission. The CEUS assessments during the drainage process were essential in ensuring accurate catheter placement and effective monitoring of HA resolution.

### 2.2. Imaging Technique

CEUS was conducted in alignment with the 2020 guidelines for liver applications [5]. The imaging utilized a GE Logiq 7 system (GE Healthcare, Milwaukee, WI, USA) equipped with a 4C convex probe. Initial B-mode scans of the liver were performed to document the size, location, and number of lesions. Following this, color Doppler imaging was conducted. The next step involved administering 2.4 mL of SonoVue contrast agent via the medial cubital vein, typically adequate for accurate diagnosis. Additional contrast was injected as required, especially in cases involving multiple lesions. The CEUS protocol employed a low mechanical index (<0.1) to preserve the microbubbles of the contrast agent [6,7]. Three primary acquisition phases were observed: the arterial phase (10–45 s), the early venous phase (45–120 s), and the late venous phase (120–180 s) [5]. After 120 s, no significant changes in the dynamic enhancement profile were noted. Lesions located deep beneath the diaphragm necessitated deep inhalation and breath-holding by patients, complicating the acquisition in certain cases. Extended acquisition was only pursued for ambiguous cases. The enhancements of the HA walls and the pus were assessed in comparison to the liver parenchyma as depicted in Figure 1. B-mode and CEUS images were independently evaluated by two radiologists to ascertain the size of the lesion and its liquid content.

### 2.3. Statistical Analysis

Statistical evaluations were performed on demographic variables such as age and sex. Echogenicity measurements for selected regions were recorded during the arterial, portal venous, and late venous phases, with the distribution characteristics displayed in boxplots that highlighted the median value at the center. The collected data showed no outliers. The plots were generated using the Python Matplotlib v3.6 package (Matplotlib, 3.6, The Matplotlib Development Team, GitHub). Given the non-normal distribution of the data, differences in echogenicity between pus and liver parenchyma, as well as between pus and the HA pouch, were analyzed using a two-sided Mann–Whitney U test. An analysis of the larger lesion sizes was conducted. The total and pus-specific areas within the HA were calculated, and the volume of pus was estimated using the formula V = 4/3πa2b, where ‘a’ represents the area of pus observed in CEUS (nonenhancing) and B-mode (hypoechoic without Doppler flow) images, and ‘b’ is the longer dimension of the HA. For Type II and III HAs, volumes of liquefied parts were aggregated from the sinuses. Lesions were categorized based on size (>6 cm and <6 cm) and the volume of the liquid component (>113 mL and <113 mL), as outlined by Lin et al. [3]. A *p*-value < 0.05 was considered to indicate statistical significance.

## 3. Results

### 3.1. Size

Lesion sizes from post-contrast CEUS sequences were as follows: the mean (SD) was 6.26 cm (3.56), the median was 5.91 cm, the range was 0.97 cm–16.04 cm, and the interquartile range (IQR) was 3.60–7.94 cm.

### 3.2. Echogenicity

The echogenicity of pus, abscess pouch, and liver parenchyma was evaluated across the arterial, early venous, and late venous phases (Figure 2).

#### 3.2.1. Arterial Phase

Pus showed significantly lower echogenicity compared to that of both the abscess pouch (U = 927.5, *p* < 0.001) and liver parenchyma (U = 845.5, *p* < 0.001).

#### 3.2.2. Early Venous Phase

The echogenicity of pus was significantly lower than that of the abscess pouch (U = 1542.0, *p* < 0.001) and liver parenchyma (U = 1122.5, *p* = 0.001).

#### 3.2.3. Late Venous Phase

Pus maintained significantly lower echogenicity compared to that of the abscess pouch (U = 1557.0, *p* < 0.001) and liver parenchyma (U = 1106.0, *p* < 0.001).

### 3.3. Volume

#### 3.3.1. Classification of HAs by Size and Volume

Since the difference in enhancement between the fluid component of HAs and their capsules was statistically significant throughout the study period, a morphological classification of HAs was performed. Additionally, the lesions were categorized based on their size, >6 cm (53 lesions) or <6 cm (59 lesions), and their volume, either >113 mL or <113 mL. Both CEUS and B-mode were consistent in determining the dimensions of HA, but they significantly differed in assessing the volume of the fluid component (Figure 3, Figure 4, Figure 5 and Figure 6).

#### 3.3.2. Comparison of HA Volume Detection Using CEUS and B-Mode in Lesions by Size

The bar chart (Figure 7) presents the percentage of abscesses with volumes >113 mL and <113 mL, evaluated using CEUS and B-mode. The data are further categorized based on lesion sizes >6 cm and <6 cm. The detection of HA containing >113 mL of liquid component for both CEUS and B-mode across lesion size categories is as follows: for lesions <6 cm, CEUS detected approximately 10% and B-mode detected 2% (*p* = 0.06). For lesions >6 cm, CEUS detected approximately 60% and B-mode detected approximately 30% (*p* = 0.0034).

## 4. Discussion

The causes of HA formation are varied and include underlying biliary tract diseases and conditions that cause peritoneal irritation, such as acute appendicitis, inflammatory bowel diseases, and diverticulosis. HA may also arise as complications from endoscopic or surgical operations on the bile ducts and liver, from interventional radiology procedures like embolization or ablation, or from trauma. The incidence of HA as complications of biliary disease is estimated at around 40% [2,4,8,9]. Without treatment, HAs frequently lead to life-threatening conditions such as sepsis, empyema, or peritonitis, typically following an HA rupture [10]. Historically, open surgical drainage was the treatment of choice for HA. However, this method carried a high risk of complications, such as hemorrhage, persistent bile drainage, or intraperitoneal pus spread, which could lead to sepsis. Currently, it is considered only for HA that do not respond to guideline-recommended management or when additional surgical interventions are necessary [2,11,12]. PCD with targeted antibiotic therapy is now considered the gold standard for treating HA. Until recently, there was an ongoing debate regarding the effectiveness comparison between PCD and percutaneous needle aspiration (PNA). Lin et al., in their meta-analysis, showed that PCD was superior to PNA in almost all respects, being associated with shorter duration of antibiotic use, faster clinical improvement, and a non-significantly increased risk of complications, which resulted in an overall higher therapeutic success [3]. Nevertheless, in some clinical cases, it is worth considering the use of the PNA. This is particularly recommended in situations where the lesions are smaller and located superficially, and placing a drain is difficult or impossible. A key issue in the case of HA is achieving a correct diagnosis. Due to its various possible morphological forms depending on the stage of the disease, the pathogen involved, or the general condition of the affected person, confidently diagnosing HA can pose some challenges [4]. Some authors emphasize the importance of differentiating hepatic abscesses from tumors that contain necrotic foci, hemorrhagic cysts, or hematoma [1,13]. Currently, the recognized forms of HA diagnosis are ultrasound and CT. Nevertheless, CT enhanced with contrast agent shows higher sensitivity than B-mode, and it also allows for preliminary differential diagnosis of the lesion [4,9,14]. As HA is a dynamic disease and its morphological form may change over time, it requires monitoring during its course. The method recognized for this purpose is contrast-enhanced CT, which involves excessive exposure of the patient to radiation and the kidneys to iodine contrast agents. A relatively new method not yet included in the latest guidelines is CEUS. According to Xie et al.’s meta-analysis, it provides diagnostic performance at a similar level to contrast-enhanced CT but can nevertheless be used in patients suffering from renal failure, without exposing the patient to radiation. Additionally, it is less expensive and offers real-time imaging, which can be utilized for interventional radiology procedures such as biopsies or drainage [15]. Radiological signs suggestive of HA include the image of a ‘shooting disc’ and the presence of gas bubbles inside the lesion. The “shooting disc” pattern of HA is characterized by the following: central HA cavity—hypoechoic or low-density area indicating fluid; radial septations—linear structures radiating outward like spokes; peripheral rim enhancement—enhanced rim on contrast CT due to inflammation; heterogeneous appearance—mixed echogenicity or densities within the HA; surrounding edema—swollen liver tissue adjacent to the HA [13,16]. The presence of gas bubbles may not be solely indicative of HA [9,13]. Serraino et al. demonstrated gas presence in 18.5% of patients in their study group [8]. Charles, in his paper, noted that, if an HA contains intracavitary air, it may hinder accurate visualization with ultrasound, suggesting that CT is preferable for this purpose [11]. In B-mode imaging, an HA containing gas bubbles can be erroneously diagnosed as a solid lesion. However, CEUS is a method that, due to its ability to correctly identify gas bubbles, can potentially replace CT in this aspect, especially for ultrasound monitoring of patients after procedures on the bile ducts, where the formation of HA and aerobilia are important complications [17]. It is important to note that the presence of gas can interfere with both B-mode and CEUS imaging, limiting the visibility and accurate assessment of the HA. This has been noted in previous studies, including those by Charles et al., which highlight the importance of careful interpretation of imaging results in the presence of gas. In our opinion, the classification of HAs created by Francicio seems to be the most appropriate for application in the diagnostic and treatment process. It enables the consideration of both solid and liquid lesions in the context of a comprehensive differential diagnosis, including those containing extensive necrotic foci [4]. Our results showed that the liquefied component of the HA does not enhance throughout all phases of the examination. This makes it possible to scan the entire liver parenchyma, allowing for the precise assessment of the number of HAs and their morphology. The lack of enhancement of the liquefied component makes it possible to select the most appropriate lesion for drainage and identify the best site for drain placement. According to our results, for HAs measuring <6 cm on B-mode imaging, only about 2% of the lesions met the designated criterion of 113 mL, while about 10% did so according to CEUS. For lesions measuring >6 cm on B-mode, approximately 30% met the criterion, whereas about 60% did so according to CEUS. These results demonstrate that CEUS allows for the correct classification of an HA for drainage with a much higher probability. Many authors point out the need to control the location of the drain within the HA. The most commonly mentioned imaging tool for this purpose is CT or fluoroscopy. However, the potential requirement to monitor the drainage progress with imaging exposes the patient to unnecessary additional radiation [11,18,19]. In our opinion, CEUS can be effectively used in this aspect. Furthermore, the assessment of the drain tip’s location can be performed immediately after its placement in the HA cavity. This eliminates the logistical difficulties associated with the need to move the patient from the ultrasound room to the CT room and reduces the aforementioned exposure to radiation. Some authors suggest that intracavitary CEUS could be promising in the management of HAs; this involves administering contrast directly into the drain shortly after its placement in the abscess cavity [20,21,22]. This enables the assessment of the interior of the HA, potentially identifying communication between its cavities in cases classified as morphological stages II and III according to the Francica system [4]. In their study, Ming et al. found that patients undergoing treatment for HAs diagnosed using intracavitary CEUS in conjunction with classic CEUS with intravenous contrast administration demonstrated a higher therapeutic success rate compared to those receiving only intravenous CEUS, with percentages of 95.56% versus 80.49%, respectively [22]. In our opinion, the simultaneous use of both contrast administration methods appears exceptionally promising. However, in the authors’ country of practice, financial constraints could pose a significant obstacle to implementing this combination. Ming et al. suggest diluting the contrast at a ratio of 1:20 [22]; however, this requires additional doses of contrast, which may pose financial challenges which would need justification. Nonetheless, we believe that this diagnostic alternative merits consideration, albeit in cases where its justification is clear or questionable. While CEUS has demonstrated enhanced visualization and measurement accuracy of HA volumes, its direct influence on therapeutic decision making remains unclear. Although our retrospective analysis indicated superior diagnostic capabilities with CEUS, the absence of direct comparative data on therapeutic outcomes limits our ability to draw definitive conclusions. Currently, we have initiated a prospective study to collect data on tracking treatment responses longitudinally. This study aims to robustly determine how CEUS may alter treatment pathways in clinical practice. Additionally, we plan to compare the clinical status of patients, laboratory tests, pathogens, and imaging findings, and correlate all these factors to provide a comprehensive analysis. While the general consensus these days is that PCD, along with targeted antibiotic therapy, is considered the gold standard for treating HA, there are calls to categorize HAs by size. Some advocate for treatment with both drainage and antibiotics, while others argue that smaller HAs can be managed with pharmacology alone [9,11,23]. Bamberger’s review suggests that all HAs measuring <5 cm in size can be effectively treated with antibiotic therapy alone, without the need for PCD. However, it is important to note that this review is based on data from 1996 to 1994, a period when imaging techniques were significantly less advanced than they are today. Therefore, measurements of lesions, particularly the fluid component within them, could be inaccurate [24]. On the other hand, Charles points out that HAs measuring <3 cm, depending on their location, may not be accessible to the operator for percutaneous treatment. This limitation may necessitate attempts at pharmacological treatment alone [11]. Doubts also arise regarding the efficacy of drainage in cases of multiple HAs in one patient. A meta-analysis by Lin et al. clearly indicated that, for HAs with a volume >113 mL, the placement of a percutaneous drain was significantly associated with overall clinical improvement in patients, reduced duration of antibiotic use, and higher overall therapeutic success [3]. Our results corroborate the observations of these authors. In every patient meeting the designated volume criterion, drainage was deemed successful: pus discharge was achieved, allowing for immediate reduction in infectious material and enabling culture sampling for antibiotic therapy selection. Additionally, CEUS demonstrated a significant advantage in evaluating the fluid component compared to B-mode, which we believe markedly improves the efficiency of patient qualification for a drainage procedure with a significant chance of success. Moreover, since the fluid component remains nonenhanced in all phases of contrast studies, while other parts of the lesion are enhanced, CEUS can be considered a convenient tool for monitoring the course of treatment. The panel of figures illustrates the process of treating an HA in the right lobe of the liver using the three imaging modalities mentioned in the manuscript. One can observe the significant differences in the ultrasound images and the similarities between the CT and CEUS images (Figure 8). PCD itself remains a subject of debate among researchers, with no current guidelines definitively dictating the course of the drainage process. The two main points of contention include the necessity of monitoring it through imaging studies, as well as the selection of the optimal timing for drain removal. The most commonly described criteria in the literature for drain removal are when the daily output is less than 10 mL and the patient shows clinical improvement. In addition, both Sari et al. and Goyal et al., in their established protocols, contend that a discharge of <10 mL should continue for a minimum of 2 consecutive days [18,20,22,25]. CT is an expensive method that exposes the patient to the harmful effects of radiation. In our opinion, CEUS is an imaging modality that, as a relatively inexpensive and non-invasive examination, can be successfully used for several purposes. It can qualify the HA for the drainage procedure, aiding in selecting the best access site to the lesion or, in the case of multiple lesions, the most suitable lesion for drainage. Furthermore, CEUS can effectively monitor the course of drainage, including monitoring residual fluid collections and potentially assessing drainage patency if it has ceased. While our study did not focus on CEUS analysis of vascular complications such as hepatic venous or portal vein thrombosis, we recognize its potential importance. Future studies should include this aspect to provide a comprehensive understanding of CEUS applications in detecting and monitoring vascular complications associated with HAs.

## 5. Conclusions

We believe that CEUS is a valuable tool for diagnosing HAs, performing therapeutic procedures, and monitoring their effects. CEUS enables precise, real-time assessment of HA morphology, including the dimensions and volume of the abscesses. HA with a liquefied volume >113 mL may qualify for effective drainage placement. Additionally, CEUS facilitates the precise selection of drainage sites, particularly in cases with multiple lesions, by identifying the most suitable location. From our perspective, CEUS can replace CT as an equally effective but less harmful method. It reduces logistical challenges associated with using multiple radiological departments and is considerably more cost-effective. Although our study supports CEUS as a safer and more cost-effective alternative to CT for initial evaluation and monitoring of HA, we emphasize the importance of comprehensive clinical evaluation. Monitoring of HA should adopt a multimodal approach that incorporates clinical findings, such as hyperthermia, and laboratory data, particularly in complex cases indicating post-treatment complications. Therefore, although CEUS can significantly reduce reliance on CT, it should be considered part of a broader diagnostic and monitoring strategy rather than a stand-alone solution.

## Figures and Tables

**Figure 1 jcm-13-05046-f001:**
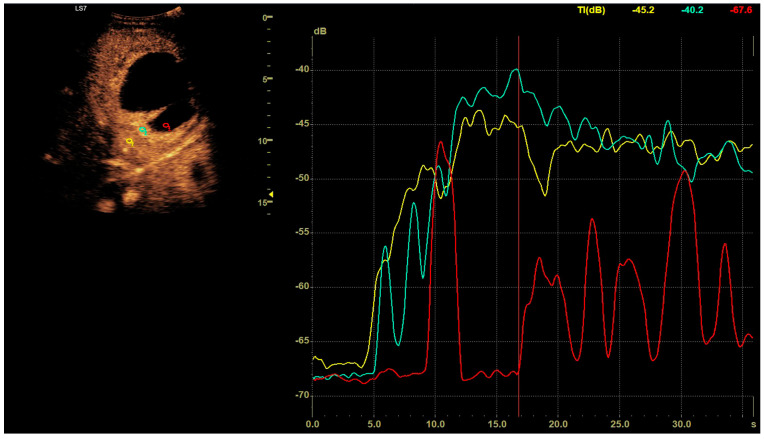
In the post–contrast CEUS assessment, regions of interest (ROIs) were designated within the lesion (colored red for pus and blue for the abscess capsule) and in the liver parenchyma (colored yellow). We tracked enhancement curves for 2–3 min, capturing data in 20–30 s intervals. Any data distorted by movement artifacts, which displaced the ROIs, were excluded from the analysis.

**Figure 2 jcm-13-05046-f002:**
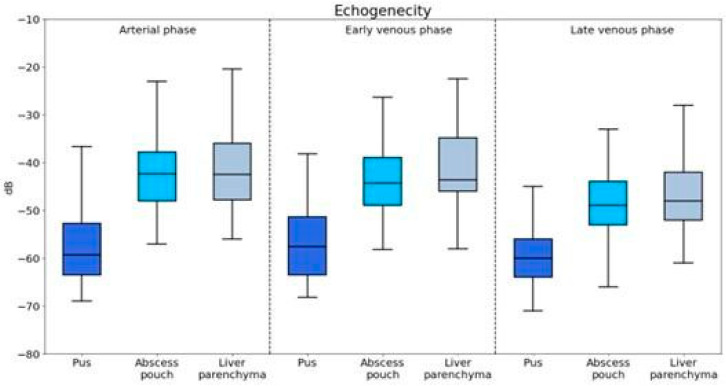
Visualization of enhancement values for different components of the abscess captured during the arterial, portal venous, and late venous imaging phases.

**Figure 3 jcm-13-05046-f003:**
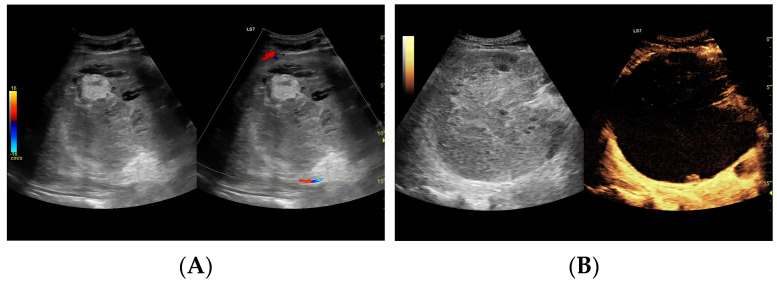
(**A**) Ultrasound and color Doppler: The transverse view shows a focal lesion involving virtually the entire liver parenchyma, identified as a type I abscess. In B-mode, the lesion appears solid and heterogeneous, with small hypoechogenic foci that can be identified as fluid fragments. Confident classification of the lesion as a hepatic abscess based on these images alone is impossible, including accurate determination of the content and location of the fluid component. (**B**) CEUS arterial phase: The enhanced abscess capsule is clearly visible, with the purulent content occupying virtually the entire lesion. There is the possibility to assess the liquefied component of the abscess with high accuracy. The total abscess was estimated to be 191.25 cm² from CEUS, while the fluid fractions were estimated at 6.51 cm² on B-mode and 166.51 cm² on CEUS, respectively, which, according to the methodology used, translates to 104.42 mL in B-mode and 2670.82 mL in CEUS, respectively. The enhancement of the abscess capsule is barely visible but similar to that of the parenchyma.

**Figure 4 jcm-13-05046-f004:**
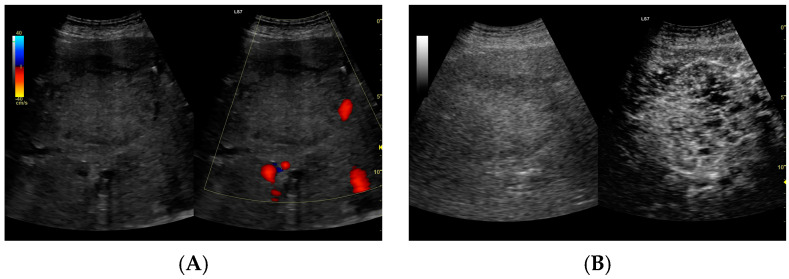
(**A**) Ultrasound and color Doppler: The transverse view shows a focal lesion in the right liver lobe, identified as type II—honeycomb. In B-mode, the lesion appears solid and heterogeneous, surrounded by a hyperechoic capsule. Confident classification of the lesion as a hepatic abscess based on these images alone is impossible, including accurate determination of the content and location of the fluid component. (**B**) CEUS arterial phase: The enhanced abscess capsule is visible, with the lesion being heterogeneously contrasted and hyperdense in relation to the liver parenchyma. Hypodense fluid areas corresponding to pus are also visible. The total abscess was estimated to be 40.71 cm² from CEUS, while the fluid fractions were estimated at 0.44 cm² on B-mode and 6.17 cm² on CEUS, respectively, which, according to the methodology used, translates to 3.23 mL in B-mode and 45.35 mL in CEUS, respectively. The enhancement of the abscess capsule is similar to that of the parenchyma, and the greatest phase of inflammatory activity has passed, indicating that the lesion is in the resolution phase of inflammation.

**Figure 5 jcm-13-05046-f005:**
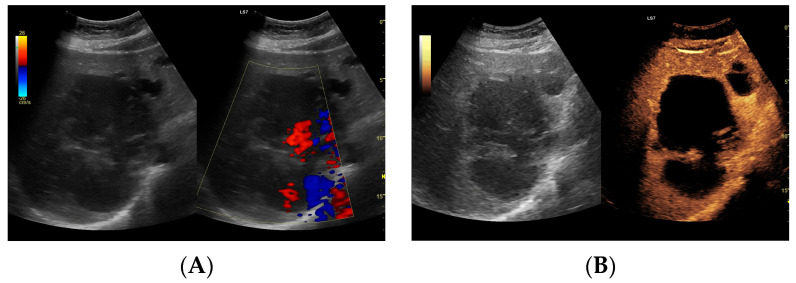
(**A**) Ultrasound and color Doppler: The transverse view shows a focal lesion in the right liver lobe, identified as a type III–, lacunar abscess, with visible septa in B-mode. In this case, it is possible to clearly classify the lesion as a type III abscess and determine its actual fluid component. (**B**) CEUS arterial phase: The enhanced abscess capsule and septa are visible. The total abscess was estimated to be 124.16 cm² from CEUS, while the fluid fractions were estimated at 57.43 cm² on B-mode and 61.55 cm² on CEUS, respectively, which, according to the methodology used, translates to 914.86 mL in B-mode and 980.49 mL in CEUS, respectively. The enhancement of the abscess capsule is slightly higher than that of the parenchyma. The lesion remains in the active phase of inflammation, although with less intensity.

**Figure 6 jcm-13-05046-f006:**
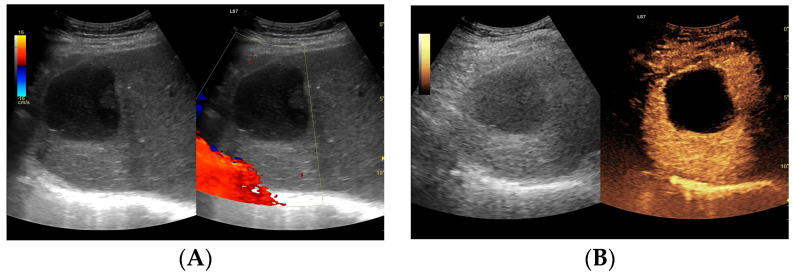
(**A**) Ultrasound and color Doppler: The transverse view shows a focal lesion in the right liver lobe, identified as a type IV–, a cystic–like abscess, with a visible capsule in B-mode. In this case, it is possible to clearly classify the lesion as a type IV abscess and determine its actual fluid component. (**B**) CEUS arterial phase: The enhanced abscess capsule is visible. The total abscess was estimated to be 25.95 cm² from CEUS. The fluid component in CEUS overlaps with the B-mode image and measures 16.15 cm², so the volume is the same in both cases at 102.71 mL. The enhancement of the abscess capsule is slightly higher than that of the parenchyma. The lesion remains in the active phase of inflammation, although with less intensity.

**Figure 7 jcm-13-05046-f007:**
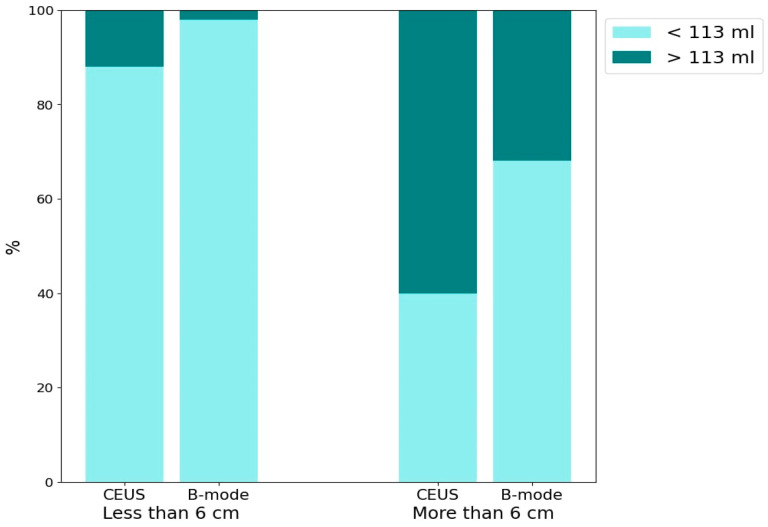
The bar chart presents the percentage distribution of two groups of abscesses categorized by liquefied component volume (<113 mL and >113 mL) using two imaging techniques: CEUS and B-mode ultrasound. The chart is divided into two main sections based on lesion size: <6 cm and >6 cm.

**Figure 8 jcm-13-05046-f008:**
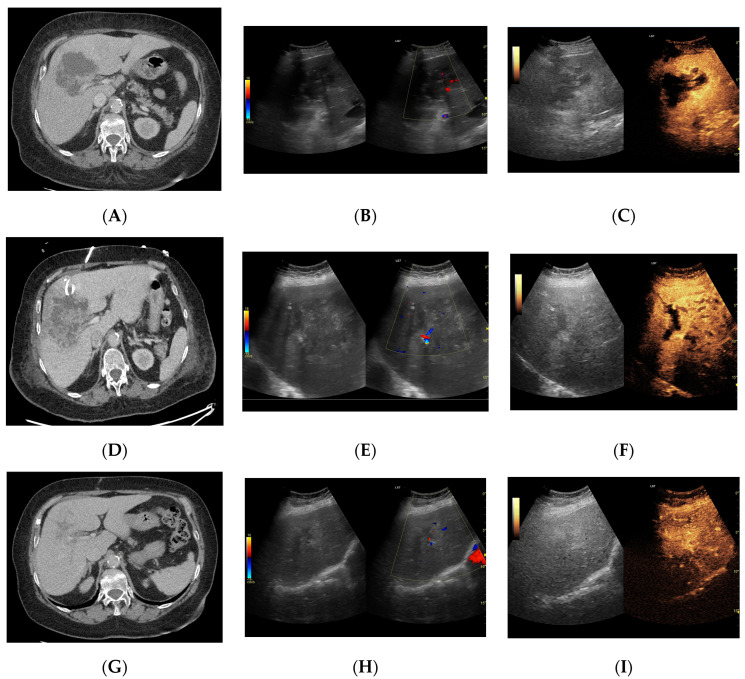
Focal liver lesion diagnosed as a type III abscess. (**A**) Post-contrast image on CT. (**B**) B-mode ultrasound: the fluid component of the lesion is difficult to define. (**C**) CEUS: the fluid component of the lesion is visible, and the patient is qualified for percutaneous drainage. (**D**) Post-contrast image on CT: Follow-up examination one week after placement of drains shows a visible reduction in the fluid component. The lesion is transitioning from type III to type II, with a visible “honeycomb” sign. (**E**) B-mode ultrasound: the fluid component of the lesion remains challenging to define. (**F**) CEUS: the fluid component of the lesion is visible, indicating regression from stage III to II. (**G**) Post-contrast image on CT: follow-up examination performed 1.5 months after placement of percutaneous drainage reveals a visible residual lesion. (**H**) B-mode ultrasound: a residual lesion is visible. (**I**) CEUS: the residual lesion shows the fluid component to be completely invisible, with a connective tissue scar present.

## Data Availability

The data presented in this study are available on request from the corresponding author.

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
