# Peer review of "Utility of Contrast-Enhanced Ultrasound in Optimizing Hepatic Abscess Treatment and Monitoring"

_jcm, 2024, doi:10.3390/jcm13175046_

Round 1

Reviewer 1 Report

Comments and Suggestions for Authors

This is a single center retrospective study investigating the utility of contrast-enhanced ultrasound (CE-US) compared to B-mode US in assessing the volume of hepatic abscess and therapeutic approach. The authors concluded that CE-US showed superior diagnostic value in evaluating the volume of fluid collection in hepatic abscess and it could replace CT. However, the following concerns need to be addressed:

Comments

1.        This study simply showed the easier assessment of fluid volume of hepatic abscess in patients who already diagnosed using CT scan, however the effect of CE-US for therapeutic approach is still unclear because they showed no data regarding the outcome after treatment. The authors should discuss how CE-US could change the therapeutic approach based on their data.

2.        The volume of hepatic abscess they evaluated in this study was estimated only by US and whether if the estimated volume was correct was not confirmed. It should be compared with the CT findings.

3.        The authors should clarify how they set their cut-off value for hepatic abscess size (6cm) and volume (113mL) for analysis.

4.        The authors concluded that CE-US could replace CT for monitoring hepatic abscess, however it would not be appropriate because monitoring hepatic abscess is performed by not only imaging findings but also other clinical findings (hyperthermia, laboratory data…) and if the post-treatment course was unfavorable CT scan is essential for considering the new hepatic abscess.

Author Response

Dear Reviewer,
Thank you very much for your extremely valuable comments. We have made every effort to
address all of your suggestions in our revised manuscript. Additionally, for the sake of clarity,
we have provided specific responses to each of your comments directly below them.
The changes in the submitted manuscript are combined with the corrections based on the
suggestions from Reviewer 2 and the editorial office.
Best regards,
1. This study simply showed the easier assessment of fluid volume of hepatic abscess in patients
who already diagnosed using CT scan, however the effect of CE-US for therapeutic approach
is still unclear because they showed no data regarding the outcome after treatment. The
authors should discuss how CE-US could change the therapeutic approach based on their data.
The following paragraph was located in the discussion section.
“While CEUS has demonstrated enhanced visualization and measurement accuracy of HA
volumes, its direct influence on therapeutic decision-making remains unclear. Although our
retrospective analysis indicated superior diagnostic capabilities with CEUS, the absence of
direct comparative data on therapeutic outcomes limits our ability to draw definitive
conclusions. Currently, we have initiated a prospective study to collect data on tracking
treatment responses longitudinally. This study aims to robustly determine how CEUS may
alter treatment pathways in clinical practice.”
2. The volume of hepatic abscess they evaluated in this study was estimated only by US and
whether if the estimated volume was correct was not confirmed. It should be compared with
the CT findings.
Every patient had initial concurrent CT and CEUS examinations, and the results were
consistent. However, in subsequent follow-up examinations, ultrasound was preferred as a
non-invasive technique. The manuscript in which we compared CEUS with CT to evaluate the
volume of HA is still awaiting review in another journal that is not open access. Based on the
results of this comparison, we proceeded to create the current manuscript.
3. The authors should clarify how they set their cut-off value for hepatic abscess size (6cm) and
volume (113mL) for analysis.
The following paragraph was located in the materials and methods section.
„Cut-off values for HA dimensions (6 cm) and volume (113 mL) were derived from a metaanalysis by Lin et al., providing a basis for differentiating between treatment approaches.”
4. The authors concluded that CE-US could replace CT for monitoring hepatic abscess, however
it would not be appropriate because monitoring hepatic abscess is performed by not only
imaging findings but also other clinical findings (hyperthermia, laboratory data…) and if the
post-treatment course was unfavorable CT scan is essential for considering the new hepatic
abscess.
The conclusion section was appropriately revised. New conclusions section:
We believe that CEUS is a valuable tool for diagnosing HA, performing therapeutic procedures, and monitoring their effects. CEUS enables precise, real-time assessment of HA morphology, including the dimensions and volume of the abscesses. HA with a liquefied volume >113 ml may qualify for effective drainage placement. Additionally, CEUS facilitates the precise selection of drainage sites, particularly in cases with multiple lesions, by identifying the most suitable location. From our perspective, CEUS can replace CT as an equally effective but less harmful method. It reduces logistical challenges associated with using multiple radiological departments and is considerably more cost-effective. Although our study supports CEUS as a safer and more cost-effective alternative to CT for initial evaluation and monitoring of HA, we emphasize the importance of comprehensive clinical evaluation. Monitoring of HA should adopt a multimodal approach that incorporates clinical findings, such as hyperthermia, and laboratory data, particularly in complex cases indicating post-treatment complications. Therefore, although CEUS can significantly reduce reliance on CT, it should be considered part of a broader diagnostic and monitoring strategy rather than a stand-alone solution.

Reviewer 2 Report

Comments and Suggestions for Authors

This article is very interesting, and well written but it contains small problems (see below).

Major points

1)    English: To be revised.

2)    (Ls.78-79): I cannot understand what the authors want to say. 11 patients were disqualified from percutaneous drainage. There were treated with antibiotics only? Please describe the treatment for them. 28 patients underwent CEUS follow-up during drainage process. Please describe the follow-up strategy (at an interval of how many days, etc). During the drainage process. It means CEUS was performed during drainage? CEUS was performed only once at admission for other 22 patients?

3)    Results: The degree of enhancement was compared between pus, abscess capsule and liver parenchyma.

At what point of parenchyma?

   In my experience, the abscess capsule is more enhanced than the liver parenchyma in arterial phase.

But, it is not so in Fig.2.

4) (Ls.243-52): I cannot understand what the authors want to say by these lines. In my opinion, the presence of gas prevents us to observe not only B mode image but also CEUS image. I agree with Charles et al.

5) In this study, CEUS of vascular complications (hepatic venous or portal vein thrombosis, --) were not analyzed. If possible, please add this point of view.

Minor point

1)    (L.22) (L.58): --confirmed hepatic abscesses via computed tomography: on the basis of what CT findings?

Please describe it precisely.

2)    (L.60) ----clinical symptoms: pain---No cases of sepsis ( or sudden onset of shock). No asymptomatic case?

3)    (L.62) –AST, ALT was considered. How about ALPγ-GPT?

4)    (L.243) “shooting disc” Please explain it briefly. It helps the readers understand CEUS of liver abscess patients.

5)    References: The journal’s name. Please abbreviate it accurately.

Author Response

Dear Reviewer,
Thank you very much for your extremely valuable comments. We have made every effort to
address all of your suggestions in our revised manuscript. Additionally, for the sake of clarity,
we have provided specific responses to each of your comments directly below them.
The changes in the submitted manuscript are combined with the corrections based on the
suggestions from Reviewer 2 and the editorial office.
Best regards,
Major points
1) English: To be revised.
The English language has been corrected.
2) (Ls.78-79): I cannot understand what the authors want to say. 11 patients were disqualified
from percutaneous drainage. There were treated with antibiotics only? Please describe the
treatment for them. 28 patients underwent CEUS follow-up during drainage process. Please
describe the follow-up strategy (at an interval of how many days, etc). During the drainage
process. It means CEUS was performed during drainage? CEUS was performed only once at
admission for other 22 patients?
The following paragraph was located in the materials and methods section:
while 11 were disqualified due to clinical status or inaccessible HA location. 28 patients
underwent CEUS follow-up during the drainage process. 11 patients who were disqualified
from PCD due to clinical status or inaccessible HA locations were treated exclusively with
targeted antibiotic therapy. For the 28 patients who underwent PCD, CEUS follow-up was
performed at 7-day intervals, or earlier if the discharge of pus from the drain ceased, to
monitor the progression and resolution of the HA. For the remaining 22 patients, CEUS was
performed only once at admission. The CEUS assessments during the drainage process were
essential in ensuring accurate catheter placement and effective monitoring of HA resolution.
3) Results: The degree of enhancement was compared between pus, abscess capsule and liver
parenchyma.
At what point of parenchyma?
 In my experience, the abscess capsule is more enhanced than the liver parenchyma in
arterial phase.
But, it is not so in Fig.2.
We agree that the abscess capsule is typically more enhanced than the liver parenchyma in
the arterial phase. Our study group was diverse and included abscesses at various stages of
development. In cases where there was a clear visual difference in the ultrasound image, as
shown in Fig. 1, the enhancement difference was at most a few dB, which is reflected in Fig.
2. The primary aim of our study was to highlight the significant contrast in the enhancement
of the fluid component of the abscess compared to other tissues, which is evident in all phases
of the examination. Relevant comments regarding the enhancement between the parenchyma
and the abscess capsule have been included under the descriptions of the respective
morphological stages in the figures within the Results section. For example, Fig. 1 shows a
fully active inflammatory phase, whereas Fig. 4 depicts the resolving phase. Figs. 5 and 6
show an active phase but with reduced intensity due to antibiotic treatment.
3) (Ls.243-52): I cannot understand what the authors want to say by these lines. In my opinion,
the presence of gas prevents us to observe not only B mode image but also CEUS image. I
agree with Charles et al.
The passage of the text to which the comment relates has been rewritten for better
understanding and readability.
4) In this study, CEUS of vascular complications (hepatic venous or portal vein thrombosis, --)
were not analyzed. If possible, please add this point of view.
The paragraph has been added to the end of the discussion section.
While our study did not focus on CEUS analysis of vascular complications such as hepatic
venous or portal vein thrombosis, we recognize its potential importance. Future studies should
include this aspect to provide a comprehensive understanding of CEUS applications in
detecting and monitoring vascular complications associated with HA.
Minor point
1) (L.22) (L.58): --confirmed hepatic abscesses via computed tomography: on the basis of what
CT findings?
Please describe it precisely.
The following paragraph was located in the materials and methods section:
HA were confirmed via CT based on the presence of hypodense lesions with peripheral
enhancement and central non-enhancing areas indicative of pus. Additional findings included
irregular HA borders and the presence of gas within the HA cavity.
2) (L.60) ----clinical symptoms: pain---No cases of sepsis ( or sudden onset of shock). No
asymptomatic case?
The following paragraph was located in the materials and methods section:
In our study, no cases of sepsis or sudden onset of shock were observed. All patients exhibited at
least one of the following clinical symptoms: pain in the right upper quadrant, jaundice,
tenderness, fever, nausea, weight loss, or night sweats. There were no asymptomatic cases
included in the study.
3) (L.62) –AST, ALT was considered. How about ALPγ-GPT?
Due to the volume of the work and the focus on the radiological aspect, we did not
specifically consider the patients' clinical conditions and laboratory results, or their changes
during the treatment process. These factors served only as eligibility criteria. We have
initiated a prospective study on this topic to further investigate and address the relationship
between radiological findings, patients' clinical conditions, and laboratory results throughout
the treatment process.
The following paragraph was located in the discussion section:
While CEUS has demonstrated enhanced visualization and measurement accuracy of HA
volumes, its direct influence on therapeutic decision-making remains unclear. Although our
retrospective analysis indicated superior diagnostic capabilities with CEUS, the absence of direct
comparative data on therapeutic outcomes limits our ability to draw definitive conclusions.
Currently, we have initiated a prospective study to collect data on tracking treatment responses
longitudinally. This study aims to robustly determine how CEUS may alter treatment pathways in
clinical practice. Additionally, we plan to compare the clinical status of patients, laboratory tests,
pathogens, and imaging findings, and correlate all these factors to provide a comprehensive
analysis
5) (L.243) “shooting disc” Please explain it briefly. It helps the readers understand CEUS of
liver abscess patients.
The term was explained in parentheses after it was cited in the discussion section.
Radiological signs suggestive of HA include the image of a 'shooting disc' and the presence of
gas bubbles inside the lesion. The "shooting disc" pattern of HA is characterized by: central
HA cavity: hypoechoic or low-density area indicating fluid, radial septations: linear structures
radiating outward like spokes, peripheral rim enhancement: enhanced rim on contrast CT due
to inflammation, heterogeneous appearance: mixed echogenicity or densities within the HA,
surrounding edema: Swollen liver tissue adjacent to the HA
6) References: The journal’s name. Please abbreviate it accurately.
The references have been revised.
With reference to the comments of the second reviewer and your comments, a significantly
rewritten subsection on patients in the materials section has been created.
